# Reinforcement Learning for Machine Learning Engineering Agents

**Sherry Yang,**[1,2]   **Joy He-Yueya**[2]   **Percy Liang**[2]
[1]New York University   [2]Stanford University
sherryyang@nyu.edu

## Abstract

Machine learning engineering (MLE) has a clear objective: Given an MLE task and a verifier (e.g., performance on some held-out data), what is the most effective way to utilize compute to achieve the best performance for the given task? Existing language model (LM) agents rely on prompting frontier LMs and accumulating experience non-parametrically by storing and retrieving experience through agent scaffolds and test-time compute. In this paper, we show that in environments such as MLE where a good verifier is available, adapting the LM parameters through gradient updates can be more effective in utilizing compute and agent's experience. Specifically, we show that agents backed by weaker models that improve via reinforcement learning (RL) can eventually outperform agents backed by much larger, but static models for a given MLE task. We identify two major challenges with RL in this setting. First, actions can take a variable amount of time (e.g., executing code for different solutions), which leads to asynchronous policy gradient updates that favor faster but suboptimal solutions. We propose *duration-aware gradient updates* in a distributed asynchronous RL framework to amplify high-cost but high-reward actions. Second, using performance on the held-out data as a reward for MLE provides limited feedback. A program that's nearly correct is treated the same as one that fails entirely (e.g., during data loading). We propose *environment instrumentation* to offer verifiable *partial credit*, using a separate, static language model to insert print statement to an existing program. Our experiments suggest that a small LM (Qwen2.5-3B) adapted with RL, when given enough compute, can solve an MLE task better than prompting a frontier model (Claude-3.5-Sonnet) with the state-of-the-art agent scaffold (AIDE) by an average of 22% across 12 Kaggle tasks[1].

## 1 Introduction

Machine learning engineering (MLE) aims to answer the question: *Given an MLE task, what is the most effective way to utilize compute to achieve the best performance on the held-out data?* Existing MLE agents rely on prompting frontier LMs with agent scaffolds (Chan et al., 2024), accumulating experience non-parametrically (e.g., by storing and retrieving previous experience) (Jiang et al., 2025). While this approach of scaling up test-time compute (Snell et al., 2024; Wu et al., 2024) can allow an agent to search for better solutions, the agent's *fundamental behavior* does not change drastically without gradient updates, which wastes valuable experience from running costly ML experiments. As shown in Figure 1, running the best agent scaffold according to MLEBench (Chan et al., 2024) with Claude3.5-Sonnet for days leads to only slightly better best solutions.

A natural approach to improving an LM agent given past experience for an MLE task is to adapt its parameters through gradient updates using reinforcement learning (RL) (Sutton et al., 1998). However, agentic settings pose additional challenges to RL. First, action execution of an MLE agent can take a variable amount of time (e.g., training different ML models), which leads to asynchronous policy gradient updates that favor faster but suboptimal solutions. To overcome this challenge, we propose *duration-aware* gradient updates during distributed asynchronous RL training to balance gradient updates to faster and slower actions. With duration-aware updates, we observe that an agent stops favoring faster solutions, achieving better performance in the long run.

---

[1]Code available at https://github.com/sherryy/rl4mle

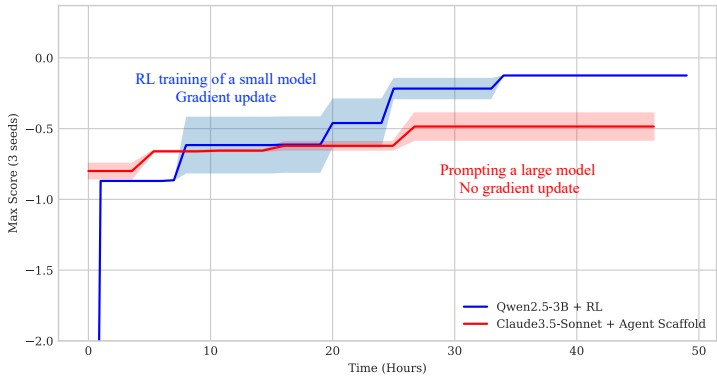

Figure 1: Performing gradient update with RL on Qwen2.5-3B (blue) is more effective in improving the best task performance than prompting Claude3.5-Sonnet with the best agent scaffold (red) on the "leaf-classification" task from MLEBench (Chan et al., 2024).

The second challenge of RL for MLE agents is that, while performance on the held-out data serves as a verifiable reward, they offer limited feedback, treating a nearly correct program (e.g., failed during writing solution to the correct location) the same as one that fails entirely (e.g., during data loading). Such limited feedback hinders the progress of RL, and further fails to capture if an agent uses ML at all to solve a problem that is meant to be solved by ML. To address limited feedback, we propose *environment instrumentation* to offer *partial credit* to intermediate steps of completing an ML task (e.g., loading the data, building and training a model). We implement environment instrumentation by using a static copy of the original LM to insert print statements in the code generated by the agent, the execution of which provides partial credit. We observe that partial credit can gradually guide the agent away from making trivial mistakes (e.g., import errors, failures to load data) and towards improving ML techniques (e.g., feature engineering and hyperparameter choices).

Across a set of 12 challenging MLE tasks, we show that adapting the parameter of a small agent (Qwen2.5-3B) through RL can eventually outperform prompting a frontier model (Claude-3.5-Sonnet) with the state-of-the-art agent scaffold (AIDE) for 8/12 tasks, achieving an average improvement over the frontier model by 22%. Our results suggest that future MLE agents should learn to balance the compute spent across inference, interaction (action execution), and performing gradient updates, as opposed to only focusing on scaling inference-time compute and interactions.

## 2 BACKGROUND

In this section, we provide relevant notations and define key learning objectives. We further discuss a few challenges of running standard RL algorithms in agentic settings.

**MLE Agent in a Markov Decision Process (MDP).** We consider a Markov Decision Process (MDP) (Puterman, 2014) represented by a tuple $\langle \mathcal{S}, \mathcal{A}, \mathcal{R}, \mathcal{P}, \mu \rangle$, consisting of a state space $\mathcal{S}$, an action space $\mathcal{A}$, a reward function $\mathcal{R} : \mathcal{S} \times \mathcal{A} \to \mathbb{R}$, a state transition probability function $\mathcal{P} : \mathcal{S} \times \mathcal{A} \to \Delta(\mathcal{S})$, and an initial state distribution $\mu \in \Delta(\mathcal{S})$. A policy $\pi : \mathcal{S} \to \Delta(\mathcal{A})$ interacts with the environment, starting from an initial state $s_0 \sim \mu$. At each interactive step $k \geq 0$, an action $a_k \sim \pi(s_k)$ is sampled from the policy and applied to the environment. The environment then transitions into the next state $s_{k+1} \sim \mathcal{P}(\cdot|s_k, a_k)$ while a scalar reward $\mathcal{R}(s_k, a_k)$ is produced. Reinforcement Learning (RL) aims to find a policy $\pi$ that maximizes the expected future rewards:

$$J(\pi) = E_{\pi, \mu, \mathcal{P}} \left[ \sum_{k=0}^{K} \mathcal{R}(s_k, a_k) \right], \tag{1}$$

where $K$ is the total number of steps. Standard RL algorithms such as policy gradient (Williams, 1992; Schulman et al., 2017) can be applied to learn the policy update rule by estimating

$$\nabla J(\pi_\theta) = E_{\pi, \mu, \mathcal{P}} \left[ \sum_{k=0}^{K} \nabla_\theta \log \pi_\theta(a_k|s_k) \hat{A}(s_k, a_k) \right], \tag{2}$$

where $\hat{A}(s_k, a_k)$ is some advantage function which can be separately estimated (e.g., by Monte-Carlo returns from $\pi$ (Williams, 1992)).

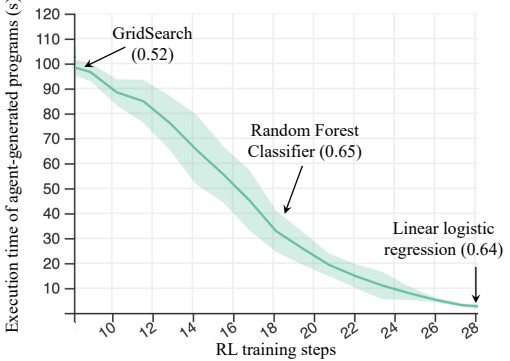

```
1  import pandas as pd
2
3  test_df = pd.read_csv(
4    "/input/test.csv")
5
6  def jaccard(str1, str2):
7    a = set(str1.lower().split())
8    b = set(str2.lower().split())
9    c = a.intersection(b)
10   return float(len(c)) / (len(a) + len(b) -
        len(c))
11
12 for i, row in test_df.iterrows():
13   phrase = max(row['text'].split(), key=
        lambda x: jaccard(x, row['text']))
14
```

Figure 2: **Execution time of agent-generated programs** (averaged across 128 samples) decreases drastically as RL training progresses. Plot produced by running distributed RL (Sheng et al., 2024) on the `random-acts-of-pizza` task (text classification). The final solution converges to a fast but suboptimal solution (achieving score 0.64) of using linear logistic regression that takes less than 1 second to execute, as opposed to other solutions with better performance (0.65) but takes longer to run.

Figure 3: **Suboptimal convergence** due to limited feedback. In a task of extracting sentiment-relevant phrases from tweets (`tweet-sentiment-extraction`), the agent converged to a suboptimal solution of directly coding the Jaccard similarity and search for the best phrase in the test input, bypassing ML completely. This demonstrates how sparse rewards can lead to an agent exploiting evaluation metrics rather than learning desired behaviors.

In the MLE agent setting, $\mathcal{S}$ captures input to the agent, including problem description, datasets, and any experiment history. $\mathcal{A}$ captures solutions generated by an agent, including high-level plans (e.g., which family of models to use) and low-level code. $\mathcal{P}$ captures any potential output from the environment during action execution (e.g., error messages, printed training losses). Note that $\mathcal{P}$ is stochastic, as the outcome of training ML models (initialized with random weights) is non-deterministic, and a good MLE agent should learn to perform well despite such stochasticity. An agent can generate a solution ($k = 1$) followed by subsequent debugging or improving steps ($k > 1$). $\mathcal{R}$ captures rewards from the environment, such as performance of the ML solution on the held-out data.

**Challenges of RL for MLE Agents.** For efficiency reasons (Thrun, 1992; Kakade, 2003), many RL training frameworks implement an asynchronous distributed setup where multiple "actors" can interact with their own instances of the environment simultaneously, gathering experiences which are then sent to a "learner" for distributed RL updates (Liang et al., 2018; Hoffman et al., 2020). In agentic settings such as ML engineering, each action may take a variable amount of time to execute. As a result, running distributed RL training favors faster actions (slower actions might often time out). Moreover, time-consuming actions are sampled less frequently in a distributed training framework, leading to an uneven number of gradient updates for faster and slower actions. As shown in Figure 2, naïvely running a distributed RL framework (Sheng et al., 2024) on a text classification task leads to the agent only generating quick solutions that barely take any time to execute.

Another challenge of RL for MLE agents is the limited feedback for intermediate progress. While performance on the held-out data is a natural reward, it does not distinguish between a solution failing to load data and one that is nearly correct. Furthermore, this lack of intermediate reward can lead to an agent not using ML to solve a problem at all. For instance, in the `tweet-sentiment-extraction` task where the agent needs to extract sentiment-supporting phrases from tweets, the agent converged to a suboptimal approach of directly coding the Jaccard similarity evaluation function and search the test input for the best phrase (as shown in Figure 3), bypassing ML completely.

## 3 RL FOR MLE AGENT

In this section, we propose duration-aware gradient updates (Section 3.1) and environment instrumentation (Section 3.2) to overcome the aforementioned challenges of applying RL to MLE agents in Section 2. Moreover, the agent can further improve a previously generated solution, which can be further enforced using RL (Section 3.3). See Algorithm 1 in Appendix A.1 for the training loop of the RL agent.

Figure 4: **Proposed framework overview**. Duration-aware gradient updates re-weights the policy gradient updates according to the execution duration of an action. Environment instrumentation inserts print statements using a static LM, the execution output can then be extracted for partial credit. The agent can be further asked to improve a previous solution, the response can further be enforced with RL.

## 3.1 DURATION-AWARE GRADIENT UPDATES FOR VARIABLE-TIME ACTION EXECUTION

A naïve solution to the aforementioned problem of variable duration actions in Section 2 is to wait for all actions to finish executing before performing any policy gradient updates, but this does not utilize resources well and is not scalable as training a model for a hard ML problem can take days.

**The Issue with Variable-Duration Action Execution.** We first provide a precise illustration of the issue with variable-time action execution in distributed RL training. Define $n_x$ and $n_y$ as the number of samples of actions $x$ and $y$ collected in time $T$. Denote $\hat{A}_x$ and $\hat{A}_y$ as the average advantage estimates for actions $x$ and $y$. In a fixed training period of length $T$, we would collect approximately:

$$n_x \approx \frac{\pi(x|s) \cdot T}{\Delta t_x} \text{ samples of action } x, \qquad n_y \approx \frac{\pi(y|s) \cdot T}{\Delta t_y} \text{ samples of action } y$$

Where $\pi(x|s)$ and $\pi(y|s)$ are the probabilities of selecting actions $x$ and $y$ under the current policy. The total gradient contribution for each action would be:

$$G_x = \frac{\pi(x|s) \cdot T}{\Delta t_x} \cdot \nabla_\theta \log \pi_\theta(x|s) \cdot \hat{A}_x, \qquad G_y = \frac{\pi(y|s) \cdot T}{\Delta t_y} \cdot \nabla_\theta \log \pi_\theta(y|s) \cdot \hat{A}_y$$

Note that $G_x$ and $G_y$ are divided by $\Delta t_x$ and $\Delta t_y$, meaning faster actions (smaller $\Delta t$) contribute proportionally more to the gradient.

**Duration-Aware Gradient Updates.** To counter the frequency bias above, we propose to weight each gradient update by the action duration, which gives

$$G'_x = \pi(x|s) \cdot T \cdot \nabla_\theta \log \pi_\theta(x|s) \cdot \hat{A}_x, \qquad G'_y = \pi(y|s) \cdot T \cdot \nabla_\theta \log \pi_\theta(y|s) \cdot \hat{A}_y$$

With duration weighting, the $\Delta t$ terms cancel out, leaving each action's contribution to the gradient proportional only to its policy probability and advantage, not to its execution frequency. This ensures that actions with longer durations receive fair consideration in policy updates despite generating fewer samples in the same time period. Generalizing from this toy example to the continuous case with arbitrary action durations, we arrive at our duration-aware policy gradient update rule:

$$\nabla_\theta J(\pi_\theta) = E_{\pi, \mu, \mathcal{P}} \left[ \sum_{k=0}^{K} \Delta t_k \cdot \nabla_\theta \log \pi_\theta(a_k|s_k) \cdot \hat{A}(s_k, a_k) \right] \tag{3}$$

Where $\Delta t_k$ is the execution duration of action $a_k$ taken at state $s_k$. This formulation ensures that in expectation, the contribution of each action to the policy gradient is proportional to its true value, regardless of how frequently it is sampled due to varying execution times. In practice, we rescale $\Delta t_k$ by the average execution time in the batch to avoid overly large gradient updates. See Appendix A.2 for the proof of the unbiased policy gradient estimate with duration-aware gradient in a distributed RL setting.

## 3.2 ENVIRONMENT INSTRUMENTATION FOR PARTIAL CREDIT

**Environment Instrumentation.** To overcome the challenge of sparse reward in MLE, we propose to introduce *partial credit* so that generated programs that fail in the beginning (e.g., during data loading) will receive less partial credit than programs that is almost correct (e.g., failed at saving output to the correct location). To avoid making too much assumption about how the agent should solve a problem, we assign partial credit only based on whether a solution completed high-level procedures including importing libraries, loading data, building ML model, training the model, and running the model on the held-out data. We propose to use another static copy of the original LM to instrument the code generated by the agent by inserting print statements into the program generated by the agent to track execution progress. The terminal output will be parsed through regex matching

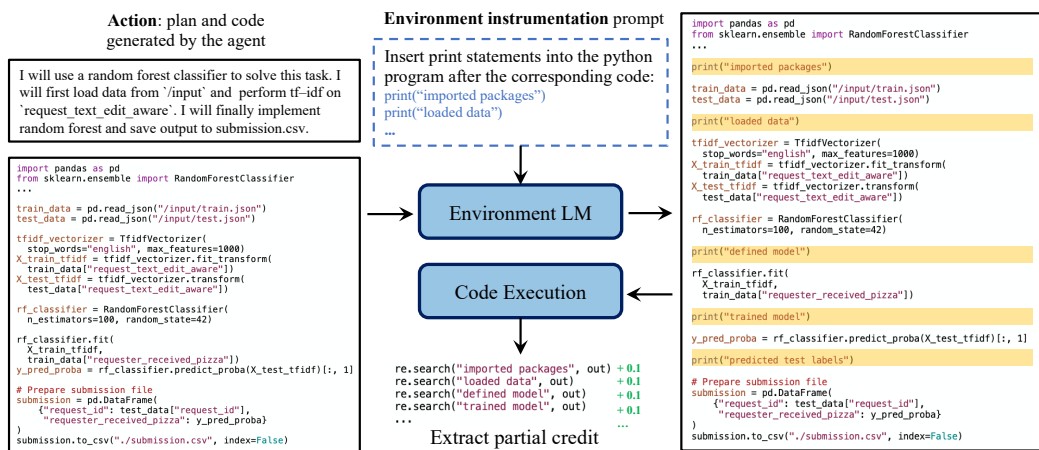

Figure 5: **Environment instrumentation overview**. Another copy of the small LM (Qwen2.5-3B) is prompted to insert print statement into the code generated by the agent. After code execution, output from the terminal is then parsed to assign partial credit by regex matching.

to provide partial credit based on whether expected print statements (e.g., "print(loaded data)") are executed, as shown in Figure 5.

**Preventing Partial Credit Hacking.** It is important to note that environment instrumentation is done by a separate, static model. If we have the same model being optimized by RL generate the print statements, we would very likely see reward hacking. By freezing the model used for instrumentation, we have observed that it largely mitigates partial credit hacking. However, instrumentation from a static model is still possible to be hacked. To further prevent partial credit hacking, we use different scales for partial credit and final reward (-10 for the worst solution, 0.1 for each of the 7 candidate print statements. If a generated programs runs without error, the submission (label for the held-out data) is graded by the grader of the environment, and true task performance is used as reward (generally between -1 and 1). As a result, the model will see much higher reward for producing valid ML solutions than to hack the print statements.

### 3.3 MULTI-STEP RL WITH SELF-IMPROVEMENT PROMPT

So far, we have discussed the setting where an agent is directly asked to generate plans and code solutions for solving MLE tasks. Next, we further explore whether we can directly instruct the agent to improve a previously generated solution. Specifically, we sample from two sets of prompts (with equal probability) to solve the problem from scratch and improve a solution generated by the previous step. We illustrate the two types of prompts in Figure 4. In the case of improving a previous solution, output of the terminal is given to the agent which includes information such as training and test accuracy (from environment instrumentation introduced in Section 3.2). We have also experimented with giving failed executions to the agent to self debug, but have noticed limited self-debugging abilities in small models. At test time, we both generate solutions from scratch and run the agent again to improve the generated solutions, and take the maximum between the two solutions (with and without explicit improvement).

### 4 EXPERIMENTS

In this section, we evaluate our proposed improvements to adapting LM agent to MLE with RL. We first discuss the evaluation setup and implementation details in Section 4.1. We then present the main evaluation results in Section 4.2, followed by ablation studies in Section 4.3.

### 4.1 EVALUATION SETUP AND IMPLEMENTATION DETAILS

**Evaluation Setup.** We perform evaluation on 12 MLE tasks from Chan et al. (2024) spanning vision, language, and tabular inputs. These tasks are selected based on whether a small model (Qwen2.5-3B) can produce a submission with valid format, because we are only interested in improving MLE, not in the general instruction following and coding abilities (which is what is required to produce a submission with the valid format). We use the grader from Chan et al. (2024) to grade the final 128 samples after RL converges and measure both the mean and the maximum performance for each run (see additional studies on the effect of sample size in Appendix D.3). We

Table 1: **Comparing RL of a small model to prompting large models** across 12 tasks from MLEBench. RL results are best scores among 128 samples after RL has converged. Baseline results are from runs in Chan et al. (2024), produced by prompting frontier models using AIDE agent scaffolds and continuing running for 24 or 100 hours. Numbers shown are mean and standard error across 3 runs. All except for the last column use the AIDE agent scaffold. ↑ denotes the higher the score the better. N/A denotes no valid submissions were available. RL of a small model achieves the best final performance on 8 out of 12 tasks.

| Tasks | Qwen2.5-3B | Llama3.1-405B | Claude3.5-Sonn | GPT-4o-100hrs | Qwen2.5-3B RL |
|---|---|---|---|---|---|
| detecting-insults-in-social-commentary (↑) | 0.870 +/- 0.009 | N/A | N/A | N/A | **0.895 +/- 0.001** |
| learning-agency-lab-automated-essay-scoring-2 (↑) | 0.331 +/- 0.018 | 0.777 +/- 0.002 | **0.794 +/- 0.008** | 0.759 +/- 0.002 | 0.746 +/- 0.002 |
| random-acts-of-pizza (↑) | 0.589 +/- 0.004 | 0.619 +/- 0.007 | 0.627 +/- 0.004 | 0.638 +/- 0.005 | **0.663 +/- 0.011** |
| tweet-sentiment-extraction (↑) | 0.027 +/- 0.018 | N/A | 0.448 +/- 0.251 | 0.283 +/- 0.005 | **0.596 +/- 0.002** |
| tabular-playground-series-may-2022 (↑) | 0.787 +/- 0.020 | 0.939 +/- 0.002 | 0.743 +/- 0.126 | 0.883 +/- 0.002 | **0.913 +/- 0.000** |
| tabular-playground-series-dec-2021 (↑) | 0.827 +/- 0.044 | 0.771 +/- 0.188 | 0.645 +/- 0.315 | **0.957 +/- 0.000** | 0.951 +/- 0.000 |
| us-patent-phrase-to-phrase-matching (↑) | 0.065 +/- 0.000 | N/A | **0.805 +/- 0.006** | 0.588 +/- 0.015 | 0.527 +/- 0.003 |
| plant-pathology-2020-fgvc7 (↑) | 0.628 +/- 0.058 | 0.968 +/- 0.005 | **0.990 +/- 0.002** | 0.970 +/- 0.001 | 0.970 +/- 0.004 |
| leaf-classification (↓) | 0.884 +/- 0.016 | 6.747 +/- 5.398 | 0.436 +/- 0.102 | 0.846 +/- 0.029 | **0.124 +/- 0.000** |
| nomad2018-predict-transparent-conductors (↓) | 0.178 +/- 0.045 | 0.166 +/- 0.103 | 0.083 +/- 0.020 | 0.072 +/- 0.003 | **0.059 +/- 0.000** |
| spooky-author-identification (↓) | 0.596 +/- 0.053 | 0.487 +/- 0.020 | 0.701 +/- 0.186 | 0.546 +/- 0.004 | **0.404 +/- 0.011** |
| lmsys-chatbot-arena (↓) | 11.48 +/- 0.002 | 1.269 +/- 0.051 | 2.211 +/- 0.959 | 1.451 +/- 0.035 | **1.081 +/- 0.002** |

use the scores achieved by different frontier LMs and different agent scaffolds from the original runs of MLEBench as baselines. The baselines prompt frontier models with the state-of-the-art AIDE agent scaffold, which organizes experience in a tree structure and saves the best solution seen so far for evaluation. In evaluating against different agent scaffolds, we use the results from the GPT-4o based agent running 24 hours using two additional agent scaffolds, OpenHands (Wang et al., 2024) and MLAgentBench (MLAB) (Huang et al., 2023b) (which has outperformed LangChain (Chase, 2022) and AutoGPT (Significant Gravitas)). To further understand the improvement progress of RL and prompting, we re-run the set of MLEBench experiments using Claude-3.5-Sonnet and AIDE agent scaffolding, while grading the intermediate best saved solutions, and compare that to intermediate solutions during RL, both across three runs.

**Implementation Details.** To implement RL for the Qwen model, we build on top of the distributed RL training framework in Sheng et al. (2024). We implement a set of distributed sandboxed code execution environments similar to Chan et al. (2024), where code execution takes place inside of the RL training loop as a part of the reward function implementation. To implement environment instrumentation, we load a separate copy of the original Qwen2.5-3B model (without performing any gradient updates on it) and ask the model to insert print statements before executing the code. The prompt for environment instrumentation can be found in Appendix C.3. To assign partial credit, we use reward -10 to denote solutions that fail completely (e.g., no plans or code, fail to import packages), and add 0.1 per regex match in the terminal output. If the solution is valid (according to the grader), we use the actual score from the grader as reward. We further experimented with normalizing the reward to a particular range (0 to 1) but did not observe significant difference. For tasks where the lower scores are better, we flip the signs of the scores to use as rewards. We use the Proximal Policy Gradient (PPO) (Schulman et al., 2017) algorithm (with modification of duration-aware gradient) to train the Qwen2.5-3B model for each task until reward convergence, which generally took 1-3 days depending on the task using 8 A100-40GiB GPUs. We limit the input and output length to 1024 tokens. The model is trained using a batch size of 128 and learning rate of 1e-5 (See complete set of hyperparameters in Appendix B.1).

## 4.2 EVALUATION RESULTS ON MLEBENCH

**Comparing against Different Frontier Models.** We report the mean and standard error across three runs of RL or prompting a frontier model using the AIDE scaffold in Table 1. Since the AIDE scaffold saves the best solution found through prompting, we also report the maximum among the 128 samples (See the mean scores across the 128 samples during RL in Figure 10 in Appendix D.1). Qwen2.5-3B with RL outperforms prompting a frontier LM on 8 out of the 12 tasks, and achieves an average of 22% improvement (measured by improvements or degradation over the baselines) over prompting Claude3.5-Sonnet. For the tasks where Qwen could not outperform the frontier models, we still observe significant improvement running RL over prompting the Qwen model with the AIDE agent scaffolding (As shown in Column 2 of Table 1). We note that running AIDE for longer hours (e.g., GPT-4o 100 hours as opposed to the default 24 hours for other columns) did not lead to significantly better performance, indicating that solutions other than prompting a large frontier LM is required to effectively achieve self-improvement.

| Generated plan | Generated code |
|---|---|
| To improve the previous solution and achieve a lower log loss on the test set, we can improve **Feature Engineering**: We can create additional features from the prompt and responses, such as the length of the texts, the presence of certain keywords, or the number of words. | `# Feature engineering`
`train_df['response_a_length'] = train_df['response_a'].apply(len)`
`train_df['response_b_length'] = train_df['response_b'].apply(len)`
`train_df['response_diff'] = (train_df['response_a_length'] -`
`                            train_df['response_b_length'])` |
| Additional Feature Engineering: We can use additional features such as the word count difference, and average word length difference between the two responses are created to capture more information about the responses. | `# Additional features`
`train_df['response_word_count_diff'] = train_df['response_a'].apply(`
`  lambda x: len(x.split())) - train_df['response_b'].apply(lambda x: len(x.split()))`
`train_df['response_avg_word_length_diff'] = train_df['response_a'].apply(`
`  lambda x: np.mean([len(word) for word in x.split()])) - train_df['response_b'].apply(`
`  lambda x: np.mean([len(word) for word in x.split()]))` |

Figure 6: **Qualitative examples** of improvements proposed by the agent during RL. [Top] earlier improvement proposed by the agent using difference between response length as features for preference prediction. [Bottom] later improvements proposed by the agent using additional features such as word count and average word length difference as features.

Table 2: **Comparing RL to different agent scaffolds**. Adapting a small model with RL outperforms prompting GPT-4o with different agent scaffolds on 9 out of the 12 tasks. Results for agent scaffolds are taken from Chan et al. (2024). Numbers show mean and standard error of the final performance according to the grader in Chan et al. (2024).

| Tasks | GPT-4o AIDE | GPT-4o OpenHands | GPT-4o MLAB | Qwen2.5-3B RL |
|---|---|---|---|---|
| detecting-insults-in-social-commentary (↑) | NaN | 0.867 +/- 0.017 | 0.749 +/- 0.039 | **0.895 +/- 0.001** |
| learning-agency-lab-automated-essay-scoring-2 (↑) | 0.720 +/- 0.031 | 0.681 +/- 0.010 | 0.533 +/- 0.080 | **0.746 +/- 0.002** |
| random-acts-of-pizza (↑) | 0.645 +/- 0.009 | 0.591 +/- 0.048 | 0.520 +/- 0.013 | **0.663 +/- 0.011** |
| tweet-sentiment-extraction(↑) | 0.294 +/- 0.032 | 0.415 +/- 0.008 | 0.158 +/- 0.057 | **0.596 +/- 0.002** |
| tabular-playground-series-may-2022 (↑) | 0.884 +/- 0.012 | 0.882 +/- 0.030 | 0.711 +/- 0.050 | **0.913 +/- 0.000** |
| tabular-playground-series-dec-2021 (↑) | **0.957 +/- 0.002** | 0.957 +/- 0.000 | 0.828 +/- 0.118 | 0.951 +/- 0.000 |
| us-patent-phrase-to-phrase-matching (↑) | **0.756 +/- 0.019** | 0.366 +/- 0.039 | NaN | 0.527 +/- 0.003 |
| plant-pathology-2020-fgvc7 (↑) | **0.980 +/- 0.002** | 0.680 +/- 0.113 | 0.735 +/- 0.052 | 0.970 +/- 0.004 |
| leaf-classification (↓) | 0.656 +/- 0.070 | 0.902 +/- 0.018 | 4.383 +/- 2.270 | **0.124 +/- 0.000** |
| nomad2018-predict-transparent-conductors (↓) | 0.144 +/- 0.031 | 0.183 +/- 0.120 | 0.294 +/- 0.126 | **0.059 +/- 0.000** |
| spooky-author-identification (↓) | 0.576 +/- 0.071 | 0.582 +/- 0.020 | 0.992 +/- 0.463 | **0.404 +/- 0.011** |
| lmsys-chatbot-arena (↓) | 1.323 +/- 0.147 | 1.131 +/- 0.019 | 10.324 +/- 4.509 | **1.081 +/- 0.002** |

In Figure 6, we provide example solutions that the Qwen agent came up with during RL in solving the `lmsys-chatbot-arena` task. This task requires the agent to come up with ML code and train a model to predict which responses generated by LMs a user will prefer. The Qwen agent is able to come up with various different feature engineering choices such as using the difference in response length, word count, and average word length as additional features.

**Comparing against Different Agent Scaffolds.** We now compare running RL on Qwen2.5-3B against running different agent scaffolds on GPT-4o. Table 2 shows that Qwen2.5-3B with RL outperforms prompting LMs with various agent scaffolds on 9 out of the 12 tasks, achieving an average improvement of 17.7% over the best scaffold for each tasks. We notice that the performance of prompting a frontier model does vary for each task across different agent scaffolds. For instance, AIDE achieved no valid submissions across 3 runs (score NaN) for Llama3.1, Claude3.5-Sonnet, and GPT-4o on `detecting-insults-in-social-commentary`, while other scaffolds can achieve valid solutions on the same task. Nevertheless, AIDE generally works better in achieving higher task performance compared to other agent scaffolds, suggesting that RL is a reliable way to improve performance that is agnostic to different choices of scaffolds.

**Performance Improvement over Time.** In Figure 7, we compare the max performance (across 128 samples) aggregated over time of training Qwen2.5-3B with RL against running AIDE agent scaffold using Claude-3.5-Sonnet. We observe that for many tasks such as `learning-agency-lab-automated-essay-scoring-2`, `tweet-sentiment-extraction` and `random-acts-of-pizza`, prompting the large model initially achieves much better performance than the small model. However, as RL goes on, performance of the smaller model improves more with gradient updates, eventually exceeding prompting a large model.

## 4.3 ABLATION STUDIES

We now present ablation studies on duration-aware gradient updates, environment instrumentation, and explicit self-improvement prompt.

**Effect of Duration-Aware Gradient.** We plot the average execution time (across 128 samples) during RL with and without duration-aware gradient in Figure 8. We see the overall runtime still decreases as RL progresses, since high-performing programs generally tend to be fast. However, with duration-aware gradient, the agent is able to find better solutions that take longer to execute (e.g.,

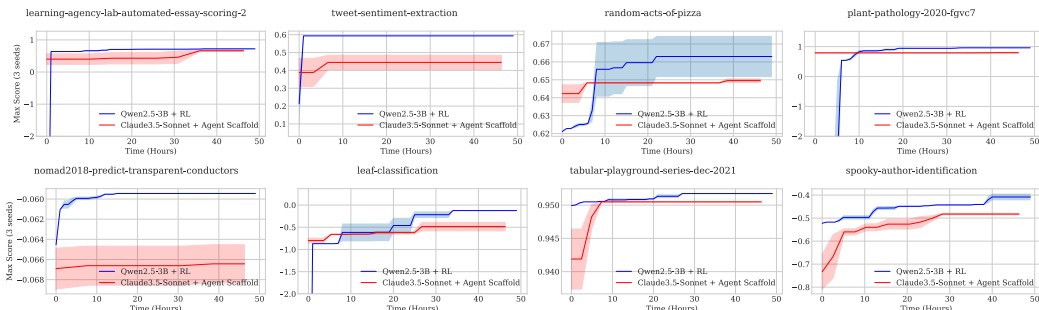

Figure 7: **The best scores achieved by the agent across time** comparing prompting a large model to adapting a small model with RL. A small model running RL starts off with low scores for many tasks, but eventually outperforms prompting a large model.

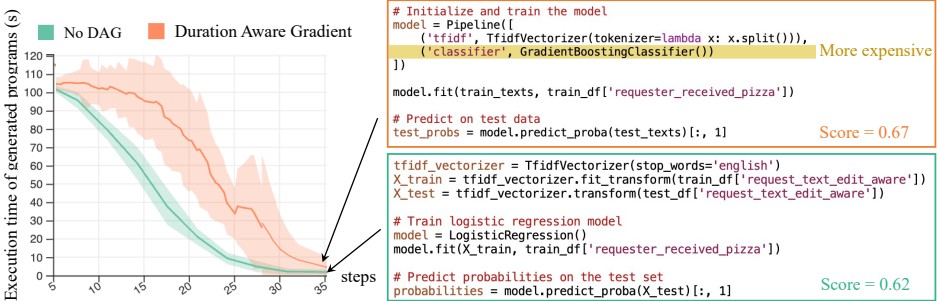

Figure 8: **Duration-aware gradient** enables the agent to explore more expensive but high-return actions by coming up with more expensive solutions such as gradient boosting, which achieves higher score than linear logistic regression. The RL agent still tends to find faster executing solutions over time.

gradient boosting), whereas without duration-aware gradient, the agent quickly converges to fast but suboptimal solution (e.g., linear logistic regression). Nevertheless, we found that the RL agent still tends to find faster executing solutions, as the average execution time still tend to decrease over time. We found that best solutions from PPO+DAG achieve an average of 3.85% improvement over vanilla PPO.

**Effect of Environment Instrumentation**  In Figure 9, we show the average task scores (across 128 samples) during RL with and without environment instrumentation. The task scores are -10 if the solutions are invalid and the actual scores otherwise (partial credits from environment instrumentation are omitted from the plot but is included in the actual reward RL optimizes). We observe that environment instrumentation leads to faster growing and faster converging average scores. The high-variance in `plant-pathology-2020-fgv7` (right most subplot) was due to one RL training run not being able to produce any valid solution due to sparse reward, which we observe more frequently when environment intrumentation is absent. We found best solutions from PPO+EI achieve 22.06% improvement over vanilla PPO.

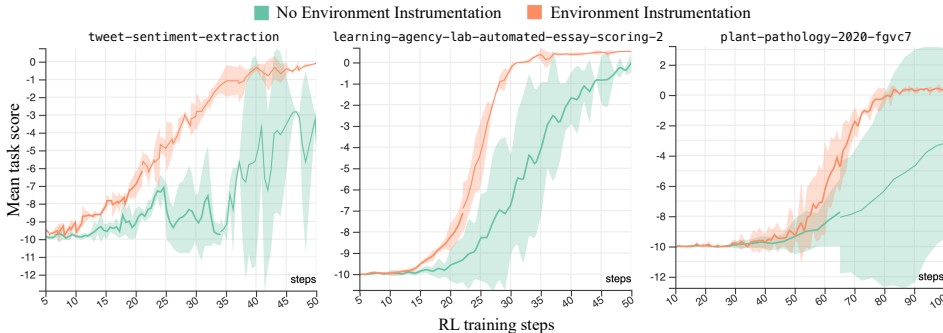

Figure 9: **Environment instrumentation ablation.** Plots show the mean task scores (excluding the partial credit from environment instrumentation) across 128 samples for 3 example tasks across RL training steps. Environment instrumentation improves RL and enables faster convergence.

**Effect of Explicit Self-Improvement Prompt.** Next, we compare explicitly asking the agent to improve a previous solution (50% of the time during RL) to only having the agent solve the task from scratch. Asking the model to improve a previous solution leads to better final performance for 10 out of 12 tasks and achieves an average improvement of 8% over coming up with solution from scratch, suggesting that RL can simultaneously improve both initial solution generation and improve the ability to improve a previously solution. See the complete performance with and without self-improvement prompt in Table 4 of Appendix D.2.

## 5 RELATED WORK

**ML engineering agents.** Many recent work has emerged for building LM agents that can solve machine learning benchmarks (Huang et al., 2023b; Tang et al., 2023; Chan et al., 2024; Li et al., 2024; Zhang et al., 2025), data science tasks (Grosnit et al., 2024; Bendinelli et al., 2025; Pricope, 2025), or help with other aspects of ML such as data preprocessing and hyperparameter optimization (Zhang et al.; Liu et al., 2024; Gu et al., 2024). Most existing work in this space has focused on prompting large frontier LMs as opposed to performing gradient updates. Existing work has used various agent scaffolds such as LangChain (Chase, 2022), AutoGPT (Significant Gravitas), OpenHands (Wang et al., 2024), and AIDE (Jiang et al., 2025) for in-context learning, and further try to improve agent performance by heuristic-based search during inference time (Liang et al., 2025). While LM agents perform better with these scaffolds, they still face the challenge of achieving improvement reliably from prompting (Huang et al., 2023a; Errica et al., 2024). We focus on RL training of smaller models instead of prompting large models.

**RL for LMs.** Since the development of both policy and value based RL algorithms (Williams, 1992; Sutton et al., 1999; Kakade, 2001; Schulman et al., 2015; 2017; Watkins & Dayan, 1992) extensively took places in simulated environment such as MuJoCo (Todorov et al., 2012) and Atari (Bellemare et al., 2013), many RL optimization frameworks (Hoffman et al., 2020; Hafner et al., 2017) make the implicit assumption is that environment interactions take up a constant amount of time. More recently, RL has been used extensively in aligning LMs to human preferences (Ouyang et al., 2022; Rafailov et al., 2023; Christiano et al., 2023; Ziegler et al., 2020), reasoning (Lee et al., 2023), and solving math (Dang & Ngo, 2025) and coding (Wei et al., 2025) problems. However, this assumption persists, as rewards are often produced by a reward model (Ouyang et al., 2022) or verifiable answers to math or coding problems (Guo et al., 2025; Wei et al., 2025). As a result, the problem of variable-time action execution has not been extensively studied. However, this problem is highly relevant in practical agentic systems such as ML engineering. As RL being extended to a broader array of agentic applications, deriving optimization frameworks that take into account the time an action takes is essential. Meanwhile, directly applying existing RL training frameworks developed for simulation settings, math, and reasoning, such as Hoffman et al. (2020); Sheng et al. (2024), results in poor agent performance.

**RL for agentic systems and interactive tasks.** Existing work has studied RL for agentic settings solving multi-step interactive tasks such as operating a unix terminal (Liu et al., 2023), booking flights (Snell et al., 2022b), controlling devices (Bai et al., 2024), negotiating price (Verma et al., 2022), navigating through the web (Zhou et al., 2024), and playing language-based games (Narasimhan et al., 2015; Snell et al., 2022a). However, most of these settings still neglect the time it takes to execute actions, and mostly leverage gamma discounting (Sutton et al., 1998) to balance the influence of future rewards and immediate rewards. These settings do not consider each turn taking a different amount of time. Additionally, sparse reward has been challenging in many agentic settings. Existing work has leveraged LMs/VLMs as process reward (Choudhury, 2025; Zhang et al., 2024; Mahan et al., 2024) to provide dense reward signals for policy evaluation and RL training (Pan et al., 2024; Venuto et al., 2024; Bai et al., 2024). However, directly using LM as reward functions can be unreliable (Son et al., 2024; Singhal et al.). We tackle sparse reward by having LM insert verifiable print statements as a form of reliable execution feedback to improve the agent through RL.

## 6 CONCLUSION

We have shown that adapting LM through RL is better than prompting a frontier model with non-parametric memory for a given MLE task, even with a much smaller model. We have also shown that reweighting policy gradient updates based on action duration can overcome variable-duration interactions, while using an LM to perform reward instrumentation through code can mitigate sparse rewards. These findings suggest future work on MLE agents to focus more on adapting model parameters with verifiable rewards, and offer tools for resolving challenges of RL in MLE settings.

Due to limited compute and small base models available, we only demonstrated the effectiveness of the proposed methods on 12 MLE tasks, where the small model was not bottlenecked by basic instruction following capabilities. Nevertheless, this research is the first to show what RL training could achieve for MLE even with a small model and limited compute. We expect our findings to hold for more tasks with a larger and better base model.

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

# Appendix

In this appendix, we provide additional details on the method (Appendix A), additional details on the experimental setups (Appendix B), prompts to LLMs (Appendix C), and additional experimental results (Appendix D).

## A ADDITIONAL METHOD DETAILS

### A.1 ALGORITHM FOR DURATION-AWARE GRADIENT UPDATES AND ENVIRONMENT INSTRUMENTATION

---

**Algorithm 1:** Policy Gradient with Duration-Aware Gradient Updates and Environment Instrumentation

---

**Input:** An LM agent $\pi_\theta$ with policy parameters $\theta$, learning rate $\gamma$, batch size $B$, sampling multiplier $m$, total training iterations $N$, a code execution environment $\mathcal{P}$, another copy of the LM `env-inst`, a dataset containing task descriptions $\mathcal{D}$, an empty buffer for previous solutions $\mathcal{D}_{\text{prev}}$.

**for** *iteration* = 1 **to** $N$ **do**

     Sample $m \cdot B$ prompts $s \in \mathcal{S} \sim \mathcal{D} \cup \mathcal{D}_{\text{prev}}$

     Sample solution for each prompt from the policy $a \sim \pi_\theta(\cdot|s)$

     Perform environment instrumentation and execute solution $s' \sim \mathcal{P}(s, \texttt{env-inst}(a))$

     Wait until $B$ executions complete, each of which takes $\Delta t$ and emits a reward $\mathcal{R}(s, \texttt{env-inst}(a))$

     Compute duration weighted policy gradient: $\nabla_\theta J(\pi_\theta) = E\left[\Delta t \cdot \nabla_\theta \log \pi_\theta(a|s)\hat{A}(s, a)\right]$

     Update policy parameters: $\theta \leftarrow \theta - \gamma\nabla_\theta J(\pi_\theta)$

     Update previous solutions: $\mathcal{D}_{\text{prev}} \leftarrow$ self-improve prompt given $a$ and $s'$

---

### A.2 THEORETICAL JUSTIFICATION FOR DURATION-AWARE GRADIENT

In Section 3.1, we provided intuitive math to why duration-reweighted gradient can counter the frequency bias. Below, we provide the proof of the unbiasedness of the policy gradient estimate with duration-aware gradient.

Let the standard reinforcement learning objective be the expected cumulative reward, $J(\theta) = \mathbb{E}_{\tau\sim\pi_\theta}[R(\tau)]$, where $R(\tau) = \sum_{t=0}^{H-1} r_t$ is the return of a trajectory $\tau$ sampled under policy $\pi_\theta$. The Policy Gradient Theorem states that the gradient of this objective is:

$$\nabla_\theta J(\theta) = \mathbb{E}_{s\sim d^\pi, a\sim\pi_\theta(s)}\left[\nabla_\theta \log \pi_\theta(a|s)\hat{A}^{\pi_\theta}(s, a)\right]$$

where $d^\pi(s)$ is the stationary state distribution induced by $\pi_\theta$ and $\hat{A}^{\pi_\theta}(s, a)$ is a valid estimate of the advantage function. Let $p(s, a) = d^\pi(s)\pi_\theta(a|s)$ denote the true state-action visitation distribution under the policy.

In the described distributed setting with asynchronous, variable-duration actions, the learning agent does not receive samples drawn from $p(s, a)$. Instead, it receives a stream of completed transitions where actions with shorter duration $\Delta t(s, a)$ appear more frequently over a fixed time period. The probability distribution of these observed samples, which we denote $p_{obs}(s, a)$, is therefore inversely proportional to the action duration:

$$p_{obs}(s, a) \propto \frac{p(s, a)}{\Delta t(s, a)}.$$

Let $C = \left(\mathbb{E}_{(s,a)\sim p}\left[\frac{1}{\Delta t(s,a)}\right]\right)^{-1}$ be the normalization constant. Then the observation distribution is explicitly $p_{obs}(s, a) = \frac{1}{C}\frac{p(s,a)}{\Delta t(s,a)}$. A naive policy gradient estimate using samples from this biased observation distribution would not yield the correct gradient direction.

To correct for this sampling bias, we employ importance sampling. We aim to compute an expectation over the true distribution $p(s, a)$ while using samples from our observation distribution $p_{obs}(s, a)$. The required importance weight $w(s, a)$ is the ratio of the target to the proposal distribution:

$$w(s, a) = \frac{p(s, a)}{p_{obs}(s, a)} = \frac{p(s, a)}{\frac{1}{C}\frac{p(s,a)}{\Delta t(s,a)}} = C \cdot \Delta t(s, a)$$

By weighting each sample from $p_{obs}(s, a)$ by $w(s, a)$, we can recover an unbiased estimate of the true policy gradient:

$$\nabla_\theta J(\theta) = \mathbb{E}_{(s,a)\sim p}\left[\nabla_\theta \log \pi_\theta(a|s)\hat{A}^{\pi_\theta}(s, a)\right]$$

$$= \mathbb{E}_{(s,a)\sim p_{obs}}\left[w(s, a) \cdot \nabla_\theta \log \pi_\theta(a|s)\hat{A}^{\pi_\theta}(s, a)\right]$$

$$= \mathbb{E}_{(s,a)\sim p_{obs}}\left[C \cdot \Delta t(s, a) \cdot \nabla_\theta \log \pi_\theta(a|s)\hat{A}^{\pi_\theta}(s, a)\right]$$

The expectation term in the final line is precisely the quantity estimated by the duration-aware gradient update rule. The derivation shows that this estimator is proportional to the true policy gradient $\nabla_\theta J(\theta)$. The proportionality constant $C$ is positive and does not depend on $\theta$, meaning an optimization step in this direction is a valid ascent direction for the policy objective $J(\theta)$, with the constant being absorbed by the learning rate. This confirms that the duration-aware policy gradient provides an unbiased estimate of the policy gradient's direction.

## B  ADDITIONAL EXPERIMENTAL DETAILS

### B.1  HYPERPARAMETERS

Table 3: Hyperparameters for RL training of MLE agent.

| Hyperparameter | Value |
|---|---|
| max_prompt_length | 1024 |
| max_response_length | 1024 |
| train_batch_size | 128 |
| total_epochs | 100 |
| nnodes | 1 |
| n_gpus_per_node | 8 |
| actor_model_type | Qwen2.5-3B-Instruct |
| actor_enable_gradient_checkpointing | True |
| actor_ppo_mini_batch_size | 128 |
| actor_ppo_micro_batch_size | 8 |
| actor_grad_clip | 1.0 |
| actor_clip_ratio | 0.2 |
| actor_entropy_coeff | 0.001 |
| actor_ppo_epochs | 100 |
| actor_learning_rate | 1e-5 |
| reference_log_prob_micro_batch_size | 8 |
| rollout_name | vllm |
| temperature | 0.7 |
| top_k | -1 |
| top_p | 1 |
| prompt_length | 1024 |
| response_length | 1024 |
| dtype | bfloat16 |
| gpu_memory_utilization | 0.2 |
| ignore_eos | False |
| enforce_eager | True |
| tensor_model_parallel_size | 8 |
| log_prob_micro_batch_size | 8 |
| critic_optimizer | |
| critic_learning_rate | 1e-5 |
| critic_model_type | Qwen2.5-3B-Instruct |
| critic_enable_gradient_checkpointing | True |
| critic_ppo_mini_batch_size | 128 |
| critic_ppo_micro_batch_size | 8 |
| critic_forward_micro_batch_size | 8 |
| critic_ppo_epochs | 100 |
| critic_grad_clip | 1.0 |
| critic_cliprange_value | 0.5 |
| ppo_gamma | 1.0 |
| ppo_lam | 1.0 |
| adv_estimator | gae |
| ppo_kl_coef | 0.001 |

### B.2  COMPUTATIONAL RESOURCES

We train the Qwen2.5-3B-Instruct model until convergence. We train on each task using 8 A100-40GiB NVIDIA GPUs usinv the hyperparameters above. The amount of time until convergence depends on the task, which varies from 1 to 3 days.

## C   EXAMPLE PROMPTS

### C.1   PROMPTS FOR SOLVING KAGGLE CHALLENGE

---

**Example Prompt for `random-acts-of-pizza`**

<lim_start|>system You are given a machine learning task. You must solve the task by training a model and running the model on the test set to produce a submission file.<lim_end|>
<lim_start|>user First outline your proposed solution in natural language (3-5 sentences), followed by a single markdown code block (wrapped in "'). Note:

- Datasets train.json and test.json are available in '/workdir/random-acts-of-pizza/prepared/public/'.
- You MUST produce a submission file at './submission.csv' by running your model on the test split.
- Your task is to predict the probability that a request resulted in a free pizza.
- Submissions are evaluated on the ROC curve between the predicted probability and the observed outcomes.
- Your can use pre-installed packages such as: 'statsmodels', 'pandas', 'timm', 'bayesian-optimization', 'scikit-learn', 'xgboost', 'numpy', 'torch', 'torchvision', 'lightGBM', 'torch-geometric'.
- You can't access the internet so don't use any pre-trained models need downloading.
- './submission.csv' should have the following format: "' request_id,requester_received_pizza t3_i8iy4,0 t3_1mfqi0,0 etc "'
- Data snippet: -> /workdir/random-acts-of-pizza/prepared/public/test.json:
  [ "giver_username_if_known": "N/A",
  "request_id": "t3_1aw5zf",
  "request_text_edit_aware": "Basically I had unexpected expenses this month out of money and desperate for a pizza. I Have a Tera account with level 48 Beserker and the account has founder status.Its not much but only thing i have right now that i can live without. Eating is much higher on my priority list right now than playing Tera. If you don't want the account I will be happy to pay it forward to someone this friday when I get my paycheck.",
  "request_title": "[Request] Don't have much but willing to trade.",
  "requester_account_age_in_days_at_request": 165.9420949074074,
  "requester_days_since_first_post_on_raop_at_request": 0.0,
  "requester_number_of_comments_at_request": 13,
  "requester_number_of_comments_in_raop_at_request": 0,
  "requester_number_of_posts_at_request": 1,
  "requester_number_of_posts_on_raop_at_request": 0,
  "requester_number_of_subreddits_at_request": 6,
  "requester_subreddits_at_request": [
  "TeraOnline",
  "Torchlight",
  "funny",
  "pics",
  "todayilearned",
  "windowsphone" ],
  "requester_upvotes_minus_downvotes_at_request": 168,
  "requester_upvotes_plus_downvotes_at_request": 240,
  "requester_username": "VirginityCollector",
  "unix_timestamp_of_request": 1364094882.0,
  "unix_timestamp_of_request_utc": 1364091282.0 , ...
  <lim_end|> <lim_start|>assistant

---

## C.2 PROMPTS FOR SELF-IMPROVEMENT

---

### Example Prompt for Self-Improvement

<|im_start|>system
You are given a machine learning task. You must solve the task by training a model and running the model on the test set to produce a submission file.
<|im_end|>
<|im_start|>user
You have implemented a previous solution. Revise the solution to improve the performance on the test set. First outline your proposed solution in natural language (3-5 sentences), followed by a single markdown code block (wrapped in "') which implements this solution. If you reuse parts of the example code, include those sections again in your final solution. Previous solution: "'{previous_plan_code}"'
<|im_end|>
<|im_start|>assistant:

---

## C.3 PROMPT FOR ENVIRONMENT INSTRUMENTATION

---

### Environment Instrumentation

Please insert print statements in the given python script. The print statements are supposed to reflect the progress of executing a script that solves a Kaggle challenge machine learning benchmark. These print statements will be used to debug the python script so it needs to capture the progress of execution.

**Print Statements:**

- print("imported packages")
- print("loaded data")
- print("defined model")
- print("training loss:")
- print("trained model")
- print("testing loss:")
- print("predicted test labels")

**Requirements:**

- Only insert print statement AFTER an operation is actually performed (e.g., data have actually been loaded).
- Insert print statements for "training loss:" and "testing loss:" if applicable (i.e., the code actually computes training or testing losses).
- Output the entire python script after inserting print statements in a single markdown code block (wrapped in "').
- Do not modify the original python code, other than inserting print statements.

Now please insert print statements for this python script: {code}

---

# D ADDITIONAL RESULTS

## D.1 AVERAGE SCORES ACHIEVED DURING RL TRAINING

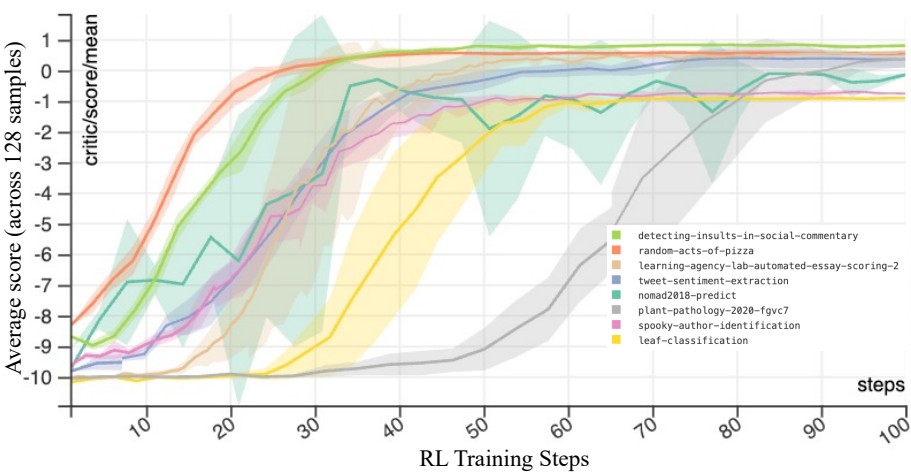

Figure 10: **Average scores** achieved during RL training for different tasks. Scores are -10 for invalid solutions and the actual score from the grader in MLEBench (Chan et al., 2024) if scores are valid. RL consistently improves average scores across tasks and across 5 seeds per task.

## D.2 FULL RESULTS FOR ABLATING SELF-IMPROVEMENT PROMPT

Table 4: **Improve a Previous Solution** leads to further improvement than only solving the task from scratch on 10 out of the 12 tasks.

| Tasks | From Scratch | Improve Previous |
|---|---|---|
| detecting-insults-in-social-commentary (↑) | **0.898 +/- 0.003** | 0.895 +/- 0.001 |
| learning-agency-lab-automated-essay-scoring-2 (↑) | 0.729 +/- 0.004 | **0.746 +/- 0.002** |
| random-acts-of-pizza (↑) | 0.643 +/- 0.004 | **0.663 +/- 0.011** |
| tweet-sentiment-extraction (↑) | 0.593 +/- 0.000 | **0.596 +/- 0.002** |
| tabular-playground-series-may-2022 (↑) | 0.902 +/- 0.000 | **0.913 +/- 0.000** |
| tabular-playground-series-dec-2021 (↑) | 0.950 +/- 0.001 | 0.951 +/- 0.000 |
| us-patent-phrase-to-phrase-matching (↑) | 0.517 +/- 0.002 | **0.527 +/- 0.003** |
| plant-pathology-2020-fgvc7 (↑) | 0.949 +/- 0.017 | **0.970 +/- 0.004** |
| leaf-classification (↓) | 0.469 +/- 0.244 | **0.124 +/- 0.000** |
| nomad2018-predict-transparent-conductors (↓) | 0.060 +/- 0.000 | **0.059 +/- 0.000** |
| spooky-author-identification (↓) | 0.448 +/- 0.000 | **0.404 +/- 0.011** |
| lmsys-chatbot-arena (↓) | 1.098 +/- 0.003 | **1.081 +/- 0.002** |

## D.3    Effect of Sample Size

MLEBench limits each run to 500 and 5000 solutions in the AIDE tree for the 24hr and 100hr run, respectively. We tried relaxing this limit to allow for a max of 1000 solutions for the 24hr run but did not observe better performance, so we used the same setting for AIDE as in MLEBench when reporting. Below is the statistics of the number of solutions for a few example tasks from running AIDE for 24hrs.

| Task | Mean | Std | Min | Max |
|---|---|---|---|---|
| detecting-insults-in-social-commentary | 445.00 | 373.08 | 62 | 1000 |
| Leaf-classification 277.00 | 377.89 | 12 | 1000 | |
| nomad2018-predict-transparent-conductors | 681.00 | 438.46 | 6 | 1000 |
| Random-acts-of-pizza | 317.00 | 306.35 | 78 | 1000 |

Table 5: Statistics of the number of solutions for a few example tasks from running AIDE for 24hrs.

As seen from Table 5, the max performance of Claude+AIDE is generally aggregated over a few hundred of solutions at most. Hence we did not want to exceed this number when aggregating over the number of samples. The choice of 128 was also partly determined by RL training, since this is generally when the base Qwen model could sample a valid submission (for RL to make progress). Below, we report the effect of sample size after RL training for a subset of the tasks in Table 6

| number of samples | detecting-insults | learning-agency | random-acts-of-pizza |
|---|---|---|---|
| 8 | 0.862 | 0.679 | 0.621 |
| 64 | 0.893 | 0.75 | 0.663 |
| 128 | 0.895 | 0.746 | 0.663 |
| 256 | 0.895 | 0.752 | 0.661 |

Table 6: Best performance under different number of samples after RL training for a subset of the tasks.

We found that the max performance is similar for sampling sizes of 64, 128, and 256. When the sampling size is reduced to 8, there is a significant performance drop.

## D.4    Alternatives to RL

We have indeed tried many different finetuning approaches to adapt the small model to a particular MLE task, including SFT, DPO, RWR (reward weighted regression), and STaR. None of these offline method worked as well as running online RL. Below is a table of running STaR (sampling a bunch of trajectories, filter high-performing trajectories to finetune the model on) and how it compared to the original Qwen model:

| Base Qwen | Filter for top 50% | Filter for top 90% |
|---|---|---|
| 0.60 +/- 0.08 | 0.59 +/- 0.12 | 0.62 +/- 0.14 |

Table 7: Performance on random-acts-of-pizza running STaR Zelikman et al. (2022) algorithm, which was uneffective for MLE.

## D.5    Additional Qualitative Examples

### D.5.1    Code Example before and after Environment Instrumentation

```python
 1  import pandas as pd
 2  from sklearn.feature_extraction.text import TfidfVectorizer
 3  from sklearn.model_selection import train_test_split
 4  from sklearn.ensemble import RandomForestClassifier
 5  from sklearn.metrics import roc_auc_score
 6  import numpy as np
 7
 8  # Load and preprocess data
 9  train_data = pd.read_json("/workdir/random-acts-of-pizza/prepared/public/train.json")
10  test_data = pd.read_json("/workdir/random-acts-of-pizza/prepared/public/test.json")
11
12  # Preprocess text data
13  tfidf_vectorizer = TfidfVectorizer(stop_words="english", max_features=1000)
14  X_train_tfidf = tfidf_vectorizer.fit_transform(train_data["request_text_edit_aware"])
15  X_test_tfidf = tfidf_vectorizer.transform(test_data["request_text_edit_aware"])
16
17  # Train a RandomForestClassifier
18  rf_classifier = RandomForestClassifier(n_estimators=100, random_state=42)
19
20  rf_classifier.fit(X_train_tfidf, train_data["requester_received_pizza"])
21
22  # Predict probabilities for the test set
23  y_pred_proba = rf_classifier.predict_proba(X_test_tfidf)[:, 1]
24
25  # Prepare submission file
26  submission = pd.DataFrame(
27      {"request_id": test_data["request_id"], "requester_received_pizza": y_pred_proba}
28  )
29  submission.to_csv("./submission.csv", index=False)
```

Figure 11: Example code before environment instrumentation for solving the `random-acts-of-pizza` Kaggle task.

```python
 1  import pandas as pd
 2  from sklearn.feature_extraction.text import TfidfVectorizer
 3  from sklearn.model_selection import train_test_split
 4  from sklearn.ensemble import RandomForestClassifier
 5  from sklearn.metrics import roc_auc_score
 6  import numpy as np
 7
 8  print("imported packages")
 9
10  # Load and preprocess data
11  train_data = pd.read_json("/workdir/random-acts-of-pizza/prepared/public/train.json")
12  test_data = pd.read_json("/workdir/random-acts-of-pizza/prepared/public/test.json")
13
14  print("loaded data")
15
16  # Preprocess text data
17  tfidf_vectorizer = TfidfVectorizer(stop_words="english", max_features=1000)
18  X_train_tfidf = tfidf_vectorizer.fit_transform(train_data["request_text_edit_aware"])
19  X_test_tfidf = tfidf_vectorizer.transform(test_data["request_text_edit_aware"])
20
21  # Train a RandomForestClassifier
22  rf_classifier = RandomForestClassifier(n_estimators=100, random_state=42)
23
24  print("defined model")
25
26  rf_classifier.fit(X_train_tfidf, train_data["requester_received_pizza"])
27
28  print("trained model")
29
30  # Predict probabilities for the test set
31  y_pred_proba = rf_classifier.predict_proba(X_test_tfidf)[:, 1]
32
33  print("predicted test labels")
34
35  # Prepare submission file
36  submission = pd.DataFrame(
37      {"request_id": test_data["request_id"], "requester_received_pizza": y_pred_proba}
38  )
39  submission.to_csv("./submission.csv", index=False)
```

Figure 12: Example code after environment instrumentation for solving the `random-acts-of-pizza` Kaggle task. All print statements are generated by environment instrumentation.

### D.5.2 EXAMPLE HIGH-PERFORMING SOLUTIONS FOUND BY THE AGENT AFTER RL TRAINING

```python
"""Plan: My proposed solution involves training a machine learning model using a combination
    of text data from request titles and descriptions, along with user-specific features such
     as account age, number of posts, etc. I will preprocess the text data using TF-IDF
    vectorization and concatenate it with the numerical features. The model will be a random
    forest classifier to handle the binary classification task of predicting whether a
    request will result in a free pizza. After training the model on the training set, I will
     fine-tune its hyperparameters using Bayesian optimization, and then use it to predict
    the probability of a free pizza for each request in the test set."""

# Import necessary libraries
import pandas as pd
from sklearn.ensemble import RandomForestClassifier
from sklearn.feature_extraction.text import TfidfVectorizer
from sklearn.model_selection import GridSearchCV
import numpy as np

# Load and preprocess data
train_df = pd.read_json('/workdir/random-acts-of-pizza/prepared/public/train.json')
test_df = pd.read_json('/workdir/random-acts-of-pizza/prepared/public/test.json')

# Feature engineering
tfidf = TfidfVectorizer(stop_words='english', max_features=1000)
tfidf.fit(train_df['request_text_edit_aware'])
train_tfidf = tfidf.transform(train_df['request_text_edit_aware'])
test_tfidf = tfidf.transform(test_df['request_text_edit_aware'])

# Combine features with user-specific features
train_features = np.hstack([train_df[['requester_account_age_in_days_at_request',
                                      'requester_days_since_first_post_on_raop_at_request',
                                      'requester_number_of_comments_at_request',
                                      'requester_number_of_comments_in_raop_at_request',
                                      'requester_number_of_posts_at_request',
                                      'requester_number_of_posts_on_raop_at_request',
                                      'requester_number_of_subreddits_at_request',
                                      'requester_upvotes_minus_downvotes_at_request',
                                      'requester_upvotes_plus_downvotes_at_request']].values,
                            train_tfidf.toarray()])
test_features = np.hstack([test_df[['requester_account_age_in_days_at_request',
                                    'requester_days_since_first_post_on_raop_at_request',
                                    'requester_number_of_comments_at_request',
                                    'requester_number_of_comments_in_raop_at_request',
                                    'requester_number_of_posts_at_request',
                                    'requester_number_of_posts_on_raop_at_request',
                                    'requester_number_of_subreddits_at_request',
                                    'requester_upvotes_minus_downvotes_at_request',
                                    'requester_upvotes_plus_downvotes_at_request']].values,
                           test_tfidf.toarray()])

# Train model
rfc = RandomForestClassifier(n_estimators=100, random_state=42)
param_grid = {'max_depth': np.arange(3, 15), 'min_samples_leaf': np.arange(2, 10)}
grid_search = GridSearchCV(rfc, param_grid, cv=3, scoring='roc_auc')
grid_search.fit(train_features, train_df['requester_received_pizza'])

# Predict probabilities on test set
test_probs = grid_search.predict_proba(test_features)[:, 1]

# Prepare submission file
submission_df = pd.DataFrame({'request_id': test_df['request_id'], 'requester_received_pizza':
    test_probs})
submission_df.to_csv('./submission.csv', index=False)
```

Figure 13: An example of high-performance (score 0.66) high cost (115 seconds) solution for `random-acts-of-pizza` achieved after RL training.

```python
"""Plan: The proposed solution involves preprocessing the train and test datasets to extract
    relevant features, training a model on the extracted features, and then predicting scores
     for the test set. For this task, I will use a transformer-based model from the 'timm'
    library to process the text data, and a lightGBM classifier for the prediction task. The
    model will be trained on the text features extracted from the train data and then
    evaluated on the test set to generate the submission file."""

import pandas as pd
import numpy as np
from sklearn.model_selection import train_test_split
from lightgbm import LGBMClassifier
from xgboost import XGBClassifier
from sklearn.feature_extraction.text import TfidfVectorizer
from tqdm.notebook import tqdm

# Load data
train_df = pd.read_csv("/workdir/learning-agency-lab-automated-essay-scoring-2/prepared/public
    /train.csv")
test_df = pd.read_csv("/workdir/learning-agency-lab-automated-essay-scoring-2/prepared/public/
    test.csv")

# Extract Text Features
tfidf_vectorizer = TfidfVectorizer()
train_tfidf = tfidf_vectorizer.fit_transform(train_df['full_text'])
test_tfidf = tfidf_vectorizer.transform(test_df['full_text'])

# Train Model
model = LGBMClassifier(n_estimators=1000, max_depth=10)
model.fit(train_tfidf, train_df['score'])

# Predict on Test Set
test_preds = model.predict(test_tfidf)

# Save Submission
submission = pd.DataFrame({
    "essay_id": test_df.essay_id,
    "score": test_preds
})
submission.to_csv("submission.csv", index=False)
```

Figure 14: An example of high-performance (score 0.73) high cost (281 seconds) solution using gradient boosting for `learning-agency-lab-automated-essay-scoring-2` achieved after RL training.

