# OpenReview forum: "Reinforcement Learning for Machine Learning Engineering Agents"
_ICLR.cc/2026/Conference — ICLR 2026 Poster_

### Official Review · Reviewer_jqr7 · 2025-10-27

**Soundness:** 3
**Presentation:** 3
**Contribution:** 2
**Rating:** 6
**Confidence:** 4

**Summary:**

This paper explores a RL framework for code generation that incorporates environment instrumentation to provide more informative rewards. By measuring both execution time and partial correctness of intermediate steps, the approach aims to make RL-based fine-tuning of code models more efficient and stable. The method is implemented on code generation benchmarks, and experimental results show that a small model trained with RL can outperform larger models or those using scaffolding, demonstrating the practical effectiveness of the proposed approach.

**Strengths:**

1. The idea of integrating execution time and partial step validity into the RL framework is innovative and interesting, providing a richer and more informative learning signal.
2. The paper is well-organized and clearly written, with logical flow and easy readability.
3. Experimental results convincingly show that a small model enhanced with RL can outperform larger models with scaffolding, validating the method’s efficiency and effectiveness.

**Weaknesses:**

1. The proposed Environment Instrumentation is conceptually interesting but appears to be an engineering optimization rather than a fundamentally new RL formulation. It would be helpful to explain why simpler signals, such as the error line number from runtime exceptions, could not serve as reward feedback.
2. The experiments lack comparisons with other RL-based code generation methods, focusing only on commercial or scaffolding-based baselines. This limits the clarity of how much the proposed method improves over existing RL techniques.
3. Minor issues: Line 245 contains a possible error, and Figure 4 seems mislabeled (the prompt content is missing and should likely correspond to Table 4).

**Questions:**

Please check my comments in Weaknesses.

---

> ### Author Response · Authors · 2025-11-21
> **Author response**
>
> Thank you for recognizing the value of this work and for the insightful questions. Please find our answers to your questions below.
>
> # Environment instrumentation is engineering as opposed to new RL formulation
>
> We agree that environment instrumentation is more of a novelty on the environment design side as opposed to the RL algorithm side. However, we think environment design is a major bottleneck for training LLM agents with RL. There are many design decisions such as how to provide rich information beyond scalar reward in the feedback and how to assign dense reward / partial credit to ensure intermediate progress. Environment instrumentation utilizes the prior that programs execute sequentially to provide dense reward. We think RL for other applications can also benefit from similar ideas, such as coding and theorem proving.
>
> # Error line number as partial credit
>
> Thanks for this suggestion. This was in fact our initial idea for partial credit before environment instrumentation, but we found a few issues with using line numbers as reward. First is that programs that have a bunch of redundant, unused imported statements ended up getting higher partial credits than programs that successfully loaded data because their line number was higher. We also had the issue where execution jumps to a helper function of a smaller line number despite being further into execution. These made the line number reward unreliable.
>
> # Comparison to other RL-based code generation methods
>
> Thanks for raising this point. The reason we didn’t compare the RL algorithm we used (PPO) to other RL algorithms is the novelty of our work is not in a fundamentally better RL algorithm, but in the implications of applying an existing RL algorithm to practical problems such as MLE, and how to resolve these practical problems (variable-time actions and sparse reward). Studying how different RL algorithms that differ in advantage estimation and off-policy handling work for MLE is not the focus of this work.
>
> That being said, we did attempt using Monte Carlo estimate of the advantage (GRPO) as opposed to generalized advantage estimation (GAE) with PPO, but it didn’t work well, as it required generating more samples simultaneously, which was not suitable for MLE where executing the samples is expensive.
>
> # Typo on line 245
>
> We indeed wanted to refer to Figure 4 on line 245 for the two types of prompt we give to the agent — coming up with a new solution and improving an existing solution (the two boxes on the left) of FIgure 4. The confusion might come from Figure 4 being on a different page than where it is referenced. We will fix this in the final paper.

---

> > ### Comment · Reviewer_jqr7 · 2025-11-25
> >
> > Thanks for your responses. I think my score is appropriate.

---

### Official Review · Reviewer_puSq · 2025-10-27

**Soundness:** 3
**Presentation:** 3
**Contribution:** 3
**Rating:** 6
**Confidence:** 3

**Summary:**

This paper explores how to best utilize compute in machine learning engineering (MLE) tasks. The paper proposes to adapt a small LM through reinforcement learning (RL), leveraging duration-aware gradient updates from the task’s verifier signal. Empirical results on 12 Kaggle-style MLE tasks show that an RL-adapted Qwen2.5-3B model outperforms a much larger static model (Claude 3.5 Sonnet) using the state-of-the-art agent scaffold (AIDE) by an average of 22%。

**Strengths:**

1. Technical contributions: Both duration-aware gradient updates and partial-credit instrumentation are conceptually clean, easy to implement, and empirically validated.
2. Strong empirical evidence: Demonstrates small adaptive models outperforming much larger static ones, which is an impactful and counterintuitive result.

**Weaknesses:**

1. Lack of detailed ablations: The effect of partial-credit feedback is not isolated in the reported experiments.
2. Generalization and robustness unclear. It is not evident whether the adapted model transfers well to unseen MLE tasks or remains stable under changing task distributions.

**Questions:**

1. Could you report performance with only duration-aware updates, and only partial-credit instrumentation, to isolate their contributions?
2. Does the RL-adapted agent transfer to unseen Kaggle tasks without additional fine-tuning?

---

> ### Author Response · Authors · 2025-11-21
> **Author response**
>
> Thank you for the insightful questions, which we answer as follows.
>
> # Isolated ablation of duration-aware gradient and environment instrumentation
>
> We did provide the isolated ablations on line 374 where “PPO+DAG achieve an average of 3.85% improvement over vanilla PPO” and on line 409 where “PPO+EI achieve 22.06% improvement over vanilla PPO.”
>
> Perhaps the reviewer was curious about the joint ablation of PPO+DAG+EI over PPO, which we didn’t report in the paper, and that improvement is 23.01%. We will be sure to include this in the final paper.
>
> # Transfers to unseen MLE tasks
>
> The focus of our evaluation is to answer the problem “given a model and some compute budget, how should an agent best solve one ML problem”, so we did not study generalization to unseen MLE tasks earlier. The options are prompting a frontier model, finetune a small model with RL, or finetune a frontier model with RL (hard to do). We pursued the second option.
>
> Generalization to unseen tasks is not the goal of the paper, but since the reviewer requested this, we also conducted an additional experiment where we used 9/12 tasks to train a single agent using RL and tested the agent on 3/12 held-out tasks. We report the cumulative best scores below:
>
> | Method |spooky-author-identification (lower is better) | random-acts-of-pizza | tweet-sentiment extraction |
> |:---|:---:|:---:|:---:|
> |Claude3.5-Sonnet + Agent Scaffold| 0.70 +/- 0.18 | 0.62 +/- 0.00 | 0.44 +/- 0.25 |
> |Zero shot Qwen2.5-3B on the 3 tasks | 0.59 +/- 0.05 | 0.58 +/- 0.00 | 0.02 +/- 0.01 |
> |Train one Qwen2.5-3B model per task for 3 tasks | 0.40 +/- 0.01 | 0.66 +/- 0.01 | 0.59 +/- 0.00
> |Train a single Qwen2.5-3B across 9/12 tasks and eval on 3/12 tasks | 0.53 +/- 0.02 | 0.61 +/ 0.01 | 0.47 +/- 0.05|
>
> We see that the RL trained model can improve the performance of the held-out tasks (although not always outperforming a frontier model).

---

> > ### Comment · Reviewer_puSq · 2025-11-25
> >
> > I appreciate the authors for their response and the additional results, which has addressed my concerns. Good luck with your submission!

---

### Official Review · Reviewer_DyMM · 2025-10-31

**Soundness:** 3
**Presentation:** 4
**Contribution:** 4
**Rating:** 8
**Confidence:** 3

**Summary:**

This is a great paper. It does RL to solve machine learning engineering tasks. From what I understood, the problems with those tasks is that when formulated as MDPs, credit assignment is difficult because the only non-trivial reward is given at the end of a trajectory when the final engineering task is solved or not. To get intermediate rewards and mitigate this problem, authors propose environment instrumentation (adding prints). They also propose duration aware gradient to favor high-cost but high-reward actions.

**Strengths:**

This paper's ideas are great and could make it a seminal paper for anyone interesting in RL for machine learning engineering.
The experiments are amazing with comprehensive ablations and multiple seeds of RL training which is rare in nowadays research.
The problem inherent to RL for MLE are well presented.

**Weaknesses:**

Here is the thing, the paper is clear as is and has very few objective weaknesses. While the propose duration aware gradients and environment instrumentation are objectively (supported by strong experiments) good contributions, I also believe that objectively the experiments do not allow to determine if RL for MLE is a real advantage. Correct me I am wrong, and I am not an expert in LLMs, but it is semms that authors only use RL for the qwen model and compare the latter to e.g. claude. To really show RL benefits for MLE, wouldn't it be better to show for a set of K models the performance differences between k_1 k_2 k_3... and k_1+RL, k_2+RL ... ?


Then, from an RL perspective, it seems to me that both duration awareness and environment instrumentation could be presented, not algorithm tricks but as MDP formulation changes: duration awareness could be added to the reward function, and prints could be added to the states and transitions? Also the asynchronous setting is not clearly motivated. It would better to focus on the fully synchronous setting for now.

**Questions:**

Could you try GAE estimatation a.k.a elligibility traces in your RL experiments for even better credit assignment?
Could you explain why asynchronicity is needed? And if not needed, would duration aware gradients still be needed?

---

> ### Author Response · Authors · 2025-11-21
> **Author response**
>
> Thank you for agreeing with the value of this work and for the positive feedback! Please find our answers to your questions below.
>
> # RL on additional models
>
> Thank you for the suggestion of running RL on more base models. We tested running RL on Llama3.1-8B-Instruct, and compared the best solutions before and after RL.
>
> |Task	|Qwen2.5-3B|	Qwen2.5-3B RL	|Improvement|
> |:---|:---:|:---:|:---:|
> |detecting-insults-in-social-commentary	|0.87|	0.895	|2.87%|
> |learning-agency-lab-automated-essay-scoring-2|	0.331|	0.746	|125.38%|
> |random-acts-of-pizza	|0.589|	0.663	|12.56%|
>
> | Tasks|	Llama3.1-8B	|Llama3.1-8B RL|	Improvement|
> |:---|:---:|:---:|:---:|
> |detecting-insults-in-social-commentary|	0.874|	0.902|	3.22%|
> |learning-agency-lab-automated-essay-scoring-2|	0.713|	0.745|	4.49%|
> |random-acts-of-pizza|	0.792	|0.790|	-0.25%|
>
> As shown in the tables above, RL improves the performance on Llama3.1-8B for “detecting-insults-in-social-commentary” and “learning-agency-lab-automated-essay-scoring-2”, but not for “random-acts-of-pizza”, and the performance gain is smaller compared to Qwen. We suspect that different base models may be more or less amenable to RL finetuning.
>
> #  MDP formulation changes
>
> We totally agree that duration awareness and environment instrumentation can indeed be presented as MDP formulation changes. We thought about the semi-MDB formulation (continuous time, discrete interventions) which could be more suited for MLE agents (e.g., learning when to terminate an unpromising ML experiment to start a new one). However, this formulation change makes it hard to utilize existing large-scale distributed RL infrastructures, but is an exciting venue of future work.
>
> We did experiment with adding time to the original reward as an additional reward, but it didn’t work well. We suspect that duration aware gradient has a similar but slightly different effect than directly modifying the reward in that it preserves the clarity of the success signal. Modifying the reward functions with time mixes two distinct objectives (correctness and speed) into a single scalar, which could make credit assignment harder.
>
> We also agree that environment instrumentation is indeed having the environment provide additional information in the next state transition. We think the novelty here is in environment design, as without instrumentation, the environment wouldn’t offer this information automatically. This environment design choice may benefit other applications such as coding and theorem proving.
>
> # Asynchronous setting
>
> We needed the asynchronous setting for efficiency reasons. In a batch of ML solutions sampled, there can be long-tail solutions that take hours to execute. Synchronously waiting for the slowest solution to finish executing is inefficient. In an asynchronous setting, we continue with gradient updates and next round of sampling. Eventually when a slow solution comes back, we use duration aware gradient and off-policy correction to utilize this slow solution to do updates, which is more efficient. It is true that duration aware gradient is not needed in a synchronous setting.
>
> # GAE estimation
>
> We indeed already use GAE for advantage estimation (Table 3 in Appendix B.1). We tried using Monte Carlo estimate of the advantage (GRPO) but it didn’t work well, as it required generating more samples simultaneously, which was not suitable for ML where executing the samples is expensive.

---

> > ### Comment · Reviewer_DyMM · 2025-11-25
> > **Thank you**
> >
> > I want to see this paper in ICLR. I increased my confidence. I trust that you will polish the paper and incorporate discussions with reviewers to the best of your ability.

---

### Official Review · Reviewer_kZcG · 2025-11-19

**Soundness:** 3
**Presentation:** 3
**Contribution:** 3
**Rating:** 6
**Confidence:** 3

**Summary:**

This paper analyzes the best way to improve performance on a MLE task with a well-defined metric and grader, specifically focusing on the benefits of prompting/scaffolding versus incremental RL. The authors explore duration-aware gradient updates and environment instrumentation / partial credit and ultimately find that a small model with RL performs best in this setting.

**Strengths:**

The authors focus on a realistic problem (machine learning engineering) and ask a very timely and relevant question (whether it's better to prompt/scaffold vs train with RL to improve performance on a task). This is a question many application businesses are asking themselves right now. The authors prove their points with practical and sound experiments that are clearly laid out. They test a variety of models, scaffolds and settings and lay out results clearly.

**Weaknesses:**

The authors focus on 12 tasks; it would be interesting to increase size (and specifically, diversity and generality) of the task set.

**Questions:**

How do these results generalize to settings beyond this 12-task set?

---

> ### Author Response · Authors · 2025-11-21
> **Author response**
>
> Thank you for agreeing with the value of this work! Please see our answer to your question about the task choice below.
>
> The choice of the tasks is based on whether a small Qwen2.5-3B model could provide a valid initial solution, which is necessary for RL to make progress. However, we want to highlight that a valid initial solution is not the same as a **good** ML solution, which is the core capability we are interested in (i.e., achieving the best performance for a given ML problem). For the rest of the tasks, the Qwen2.5-3B model was too small to follow instructions properly to understand the complex task descriptions and the dataset structures.
>
> Another way to describe our findings is that "a small LLM trained with RL produces a higher improvement on tasks where it can see valid solutions than a frontier LLM with prompt optimization". Our proposed method improves the performance of the solutions by an average of 2.76X over the base Qwen model.

---

### Comment · Area_Chair_cAxG · 2025-11-25

Dear Reviewer, thank you for reviewing for ICLR. Since the discussion deadline is coming soon, could you please take a look at the author's rebuttal, respond to their comments, and update your rating as well? Thanks!

---

### Meta-Review · Area_Chair_NPEe · 2025-12-28

**Summary:**

The four reviews are positive about this paper. Initial concerns include limited base LLM models, MDP formulation, transferability to unseen MLE, and comparison with other RL algorithms, etc. I think the authors' response adequately addresses or clarifies these concerns. Some reviewers also acknowledge that.

Thus, I recommend acceptance.

**Reviewer Concerns:**

As mentioned above.

**Reviewer Scores:**

Three reviewers already confirmed their scores.

---

### Decision · Program_Chairs · 2026-01-26

Accept (Poster)